# The Effects of Genetic Relatedness on the Preterm Infant Gut Microbiota

**DOI:** 10.3390/microorganisms9020278

**Published:** 2021-01-29

**Authors:** Shen Jean Lim, Miriam Aguilar-Lopez, Christine Wetzel, Samia V. O. Dutra, Vanessa Bray, Maureen W. Groer, Sharon M. Donovan, Thao Ho

**Affiliations:** 1College of Nursing, University of South Florida, Tampa, FL 33612, USA; jxl2581@miami.edu (S.J.L.); sozoriod@utk.edu (S.V.O.D.); mgroer@usf.edu (M.W.G.); 2Rosenstiel School of Marine and Atmospheric Science, University of Miami, Miami, FL 33149, USA; 3Division of Nutritional Sciences, University of Illinois Urbana-Champaign, Urbana, IL 61801, USA; aguilarl@illinois.edu (M.A.-L.); sdonovan@illinois.edu (S.M.D.); 4Neonatology Division, Carle Hospital, Urbana, IL 61801, USA; Chris.Wetzel@carle.com; 5College of Nursing, University of Tennessee, Knoxville, TN 37916, USA; 6College of Medicine, University of South Florida, Tampa, FL 33612, USA; vtbray@usf.edu

**Keywords:** preterm infant, gut microbiota, human milk, twins, triplets

## Abstract

The preterm infant gut microbiota is influenced by environmental, endogenous, maternal, and genetic factors. Although siblings share similar gut microbial composition, it is not known how genetic relatedness affects alpha diversity and specific taxa abundances in preterm infants. We analyzed the 16S rRNA gene content of stool samples, ≤ and >3 weeks postnatal age, and clinical data from preterm multiplets and singletons at two Neonatal Intensive Care Units (NICUs), Tampa General Hospital (TGH; FL, USA) and Carle Hospital (IL, USA). Weeks on bovine milk-based fortifier (BMF) and weight gain velocity were significant predictors of alpha diversity. Alpha diversity between siblings were significantly correlated, particularly at ≤3 weeks postnatal age and in the TGH NICU, after controlling for clinical factors. Siblings shared higher gut microbial composition similarity compared to unrelated individuals. After residualizing against clinical covariates, 30 common operational taxonomic units were correlated between siblings across time points. These belonged to the bacterial classes Actinobacteria, Bacilli, Bacteroidia, Clostridia, Erysipelotrichia, and Negativicutes. Besides the influence of BMF and weight variables on the gut microbial diversity, our study identified gut microbial similarities between siblings that suggest genetic or shared maternal and environmental effects on the preterm infant gut microbiota.

## 1. Introduction

A balanced infant gut microbiota is important for the maturation of intestinal functions [1], development of innate and adaptive immune responses [2], weight gain [3], growth [4], and even long-term health [5]. The preterm infant gut microbiota is influenced by gestational age, birth weight, and other clinical factors associated with preterm birth [6]. Compared to the term infant gut microbiota, the preterm infant gut microbiota is characterized by reduced microbial diversity, lower abundances of obligate anaerobes, dominance of potentially pathogenic *Enterobacteriaceae* (including *Citrobacter*, *Escherichia*, *Enterobacter*, *Klebsiella*, and *Serratia*, among others), and delayed colonization by commensal gut bacteria (such as *Clostridium* and *Veillonella*) typical of adult gut microbiota [4,5,6,7,8,9,10,11]. These deviations, collectively referred to as dysbiosis, are associated with increased susceptibility of infections and diseases, including necrotizing colitis (NEC) [11,12], bacteremia [13,14], late onset sepsis [15], impaired growth [4], and possibly other long-term health consequences [6].

The preterm infant gut microbiota is shaped by various therapeutic, environmental, and endogenous factors [6,16]. Therapeutic interventions include the administration of histamine-2-receptor blockers in infants [17], and, more significantly, the administration of prophylactic antibiotics in mothers [18,19] and infants [12,20]. Maternal intrapartum antibiotic use is associated with increased *Enterobacteriaceae* abundances in the gut of preterm infants [19]. Similarly, postnatal antibiotic use in preterm infants increases *Enterobacter* abundances and decreases gut microbial diversity and *Escherichia* abundances [12,20]. Thus, antibiotic-mediated shifts in gut bacterial diversity likely contribute to gut dysbiosis in preterm infants and its associated adverse health effects, including NEC, sepsis, and even death [20]. Environmental influences on preterm infant gut microbiota include effects of delivery mode [11,19], enteral feeding of formula, breast milk, and/or human milk-based fortifier (HMF) [21,22,23], and relatively stronger effects of Neonatal Intensive Care Unit (NICU) exposure [7,24]. On the other hand, endogenous influences on preterm infant gut microbiota can include maternal physiology (such as stress, chronic disease, effects from smoking, and infections) [6,16,25], infant gut physiology, and immune system responses [26].

Host genetics, previously reported to influence adult [27,28,29,30] and healthy term infant microbiota [31], can also play a role in shaping preterm infant gut microbiota [12]. Previous studies reported similarities in gut microbiota composition between preterm twins and multiplets [12,32], although sample sizes were small (<15 multiplet sets). Beyond gut compositional similarities, high-throughput sequencing has not been applied to identify specific effects of genetic relatedness on gut microbial diversity, including alpha diversity and bacterial abundances. Currently, it is also not known whether genetic similarities between preterm infant siblings are solely due to genetics or a combination of clinical factors, including antibiotics exposure, diet, maternal exposure, or NICU exposure. Larger-scale, multi-center analyses comparing gut microbial abundances between multiplets and singletons within and between centers can advance our current understanding of genetic and environmental influences on the preterm infant gut microbiota. These findings have the potential to improve clinical care of preterm neonates.

In this study, we sought to examine preterm gut infant microbiota from multiplets and singletons admitted to two different neonatal intensive care units (NICUs) located in Tampa General Hospital (TGH; FL, USA) and Carle Hospital (Carle, IL, USA). We analyzed the clinical data and microbial diversity of stool samples collected before and after three weeks postnatal age to evaluate the effects of clinical factors and genetic relatedness on gut microbial alpha and beta diversity. We used the three-week postnatal age cutoff because the gut microbiota composition of predominantly preterm, low birthweight infants was previously shown to resemble the composition of age-matched, normal birthweight infants between 18 and 21 days of life [32].

## 2. Materials and Methods

### 2.1. Enrollment and Sample Collection

After approvals from the respective Institutional Review Boards, preterm infants were enrolled from two level III NICUs, TGH and Carle NICUs, between 2012–2017 and 2016 and 2018, respectively. The TGH NICU provides one room for each infant that is equipped with a refrigerator and a milk warmer. Twins and triplets in TGH were housed in two or three separate rooms, which often shared a removable wall. On the other hand, the 48-bed-capacity Carle Hospital NICU has an open bay unit configuration with multiple pinwheel layout. Each pinwheel contains three back-to-back bedspaces and a handwashing station. Several infants share a milk warmer and a refrigerator for prepared daily feedings. There are two private rooms reserved for rooming in prior to discharge. The Carle and TGH NICUs had similar antibiotic, feeding volume, and advancement practices. However, Carle provided an exclusive human milk diet for infants born <28 weeks with the use of HMF (Prolacta Bioscience, Industry, CA, USA) and TGH NICU only provided bovine-based fortifier (BMF; Similac^®^ Human Milk Fortifier, Abbott Nutrition, Columbus, OH, USA) during the study period.

The study inclusion criteria included birth weight <1500 g or birth gestation <33 weeks. Infants with major chromosomal or congenital intestinal anomalies were excluded. Weekly infant stool samples were collected from diapers from enrollment until discharge. Samples were either placed on dry ice (Carle) or placed in a −20 °C freezer (TGH), prior to storage at −80 °C until DNA extraction and analysis. Demographic and perinatal data collected from the electronic medical records included delivery method, sex, postnatal age (days), corrected gestational age (weeks), days of antibiotic exposure, weeks on BMF/HMF milk fortifier, birth weight, current weight, weekly average weight gain velocity (g/day), and proportions of consumed mother’s own milk, formula, and donor human milk (from the week prior to each stool sample collection).

### 2.2. Sample DNA Extraction and Sequencing

Total DNA from stool specimens in the Carle cohort was extracted using the QIAamp Fast DNA Stool Mini Kit (QIAGEN, Valencia, CA, USA) with the bead-beating method previously described [33]. The V3-V4 region of the 16S rRNA gene was amplified with dual-indexed primers (Appendix A
Appendix A) and sequenced at the Roy J. Carver Biotechnology Center at the University of Illinois on the Illumina MiSeq 2 × 250 bp platform (San Diego, CA, USA). DNA extraction and library preparation protocols for the TGH stool samples were as previously published [22]. Briefly, total DNA was extracted using the QIAamp^®^ PowerFecal DNA Kit (QIAGEN, Valencia, CA, USA; previously known as MoBio PowerFecal DNA kit). The V4 region of the 16S rRNA gene was amplified using modified 515F and 806R primers [22] (Appendix A
Appendix A) and sequenced on Illumina’s MiSeq 2 × 300 bp platform. 

### 2.3. Bioinformatics and Statistics

Demultiplexed reads were quality filtered at a Q = 25 cutoff using the Trim Galore! v0.4.4, (https://github.com/FelixKrueger/TrimGalore) wrapper package, then imported into QIIME2-2019.17 [34] and denoised with DADA2 [35] into amplicon sequence variants (ASVs; Appendix A
Appendix A) without further trimming. Representative ASV sequences from the V3-V4 region (Carle cohort) and the V4 region (TGH cohort) of the 16S rRNA gene were deduplicated using CD-HIT v4.8.1 [36] at a global sequence identity threshold of 100%, retaining only the longest sequence for each ASV. Because different primer sets were used for sequencing and the ASV sequences were mapped to the same SILVA v132 [37] reference database, ASV sequences were further clustered de novo into operational taxonomic units (OTUs) at 99% identity, as recommended by Knight et al. [38]. The OTU table was filtered to remove OTUs with <10 total frequency and OTUs present in <2 samples. For phylogenetic diversity analysis, a tree containing all OTU sequences was generated using the SATé-Enabled Phylogenetic Placement (SEPP) fragment insertion method [39]. Taxonomies were assigned to OTU sequences using a naïve Bayes classifier trained on full-length 16S rRNA gene sequences from the SILVA v132 [37] database. Any OTUs with ambiguous taxonomic assignments were manually re-classified to the genus level using web blastn searches against NCBI’s nt database [40], if the best hits unanimously matched to a single genus with ≥97% identity [41]. To maintain sample size consistency in all downstream analyses, samples with missing metadata were removed from the OTU table, followed by non-singleton samples with no corresponding sibling data at each time point.

Alpha diversity metrics were calculated from the rarefied OTU table (rarefied to 1232 OTUs per sample) using QIIME2 [34]. Among the alpha diversity metrics computed from the rarefied OTU count data (*n* = 1232), Shannon’s diversity [42] was highly correlated with Pielou’s evenness (Spearman rho = 0.85, *p* < 0.001) [43], while the number of observed OTUs was significantly correlated with Faith’s phylogenetic diversity [44] (Spearman’s rho = 0.80, *p* < 0.001). For downstream analyses, we chose Shannon’s diversity [42] and Faith’s phylogenetic diversity [44] as representative alpha diversity metrics. This is because null values of Pielou’s evenness can arise from zero division [43] and because Pielou’s evenness is derived from Shannon’s diversity [42]. Compared to observed OTUs, Faith’s phylogenetic diversity carries more information because it uses phylogenetic information to estimate richness [44]. Prior to downstream analyses, alpha diversity variables and clinical variables were transformed to a near-normal distribution using R’s box cox function in the car package [45] or log-transformation (for the “days on antibiotic” variable with right-skewed distribution and zero values). Correlograms between variables before and after transformation were calculated, based on Spearman’s correlations, and visualized using R’s PerformanceAnalytics package (https://github.com/braverock/PerformanceAnalytics; accessed on 10 August 2020).

To examine associations between alpha diversity and clinical variables, linear mixed effect models were fitted with R’s lme4 [46] and nlme (https://CRAN.R-project.org/package=nlme) packages using the maximized log-likelihood method. Clinical variables were input as fixed effect predictors and alpha diversity as the response variable. Model fit was evaluated using the likelihood ratio test, Aikake Information Criterion [47], and Bayesian Information Criterion. Statistically significant predictors (*p* < 0.05) were identified using ANOVA tests. The random structure of the models was tested using various combinations of family, individual, and zygosity, keeping the fixed effect predictors and response variable constant. Based on goodness-of-fit comparisons, family was chosen as the random effect in all models. Taxonomic abundance analyses and principal components analysis (PCA) were performed using the R package ampvis2 v2.6.5 on the rarefied count table, as recommended [48]. Prior to PCA, OTUs present in <0.1% relative abundance in any sample were filtered, and the count data were transformed using the Hellinger method [49]. Shared OTU analyses were performed on unrarefied OTU counts using Venny v2.1 (https://bioinfogp.cnb.csic.es/tools/venny/). Differential abundance analysis between TGH and Carle NICU samples was performed on unrarefied counts using the default parameters of Linear Discriminant Analysis (LDA) Effect Size (LEfSe) [50]. Beta diversity was calculated from unrarefied centered-log-ratio-transformed OTU counts using the Aitchison distance, which accounts for the compositionality of OTU count data [51]. Pairwise distances between siblings and non-siblings were compared using Mann–Whitney-U tests and false-discovery-rate-adjusted *p*-values [52]. Spearman correlations of OTU abundances between siblings were calculated in R, using unrarefied OTU abundances that were residualized against clinical variables (covariates) with the umx_residualize function in the R package umx v1.9.1 (https://www.rdocumentation.org/packages/umx).

## 3. Results

### 3.1. Clinical Characteristics of Samples

Eleven twin sets, 2 triplet sets, and 32 singletons were recruited from the Carle NICU, while 22 twin sets, 4 triplet sets, and 24 singletons were recruited from the TGH NICU (Appendix A
Appendix A). From these individuals, a total of 264 stool samples were collected before and after three weeks postnatal age for this study (Table 1). Birth variables, including gestational age at delivery and weight were not significantly different between the TGH and Carle cohorts. Overall, the TGH cohort had a significantly higher proportion of male infants, and a significantly higher rate of cesarean delivery (Table 1). In the TGH cohort, the multiplets had greater birth gestational age and weight than those from Carle (Table 1). Two stool specimens, ≤ and >3 weeks postnatal age, were collected from each infant. Seven singletons from the TGH NICU were missing samples from >3 weeks postnatal age, while three twin sets from the TGH were missing samples from ≤3 weeks postnatal age. Corrected gestational ages (sum of chronological and birth gestational ages) from both NICUs were not significantly different at the time of stool collection (Table 2). One infant from the TGH cohort was diagnosed with NEC after the collection of the included stool samples. An amount of 20% of the infants in the TGH cohort were diagnosed with culture-positive sepsis. None of infants from Carle cohort developed NEC or sepsis. Exposures to antibiotics and formula were greater in the TGH cohort compared to the Carle cohort (Table 2).

### 3.2. Gut Microbial Alpha Diversity

The 16S rRNA gene sequencing of stool specimens resulted in 99.5–100% Good’s coverage values [53], indicative of comprehensive microbial community sampling. The final unrarefied OTU table contained 285 OTUs from 140 individuals, including 56 singletons, 33 twin sets, and 6 triplet sets. Alpha diversity, calculated from the OTU table rarefied to 1232 OTUs per sample, did not significantly differ between NICUs, sex, or delivery method. Across all samples, both alpha diversity metrics positively correlated with weeks on BMF, corrected gestational age, current weight, and weight gain velocity, but negatively correlated with percentage of donor human milk (Appendix A
Appendix A). Linear mixed effects modelling revealed weeks on BMF to be a significant predictor of alpha (Shannon’s and/or Faith’s phylogenetic diversity) in the Carle multiplets, Carle singletons, TGH multiplets, and TGH singletons subgroups (Figure 1). Weight gain velocity was a significant predictor of Faith’s phylogenetic diversity in these subgroups, except for TGH singletons, while current weight was a significant predictor of Shannon’s diversity in the Carle multiplets and singletons subgroups (Figure 1). Days on antibiotics also predicted Shannon’s diversity in TGH singletons, although the *p*-value was slightly above the significance threshold (*p* = 0.052; Figure 1). After residualizing against clinical variables, multiplets shared significant pairwise correlations of Faith’s phylogenetic diversity at early, but not late timepoints (Figure 2A). On the other hand, significant pairwise correlations of Shannon’s diversity were detected at both time points from the TGH, but not Carle NICU (Figure 2B).

### 3.3. Gut Microbial Composition and Beta Diversity

Stool samples from both Carle and TGH NICUs were dominated by the phyla Firmicutes (average 52% relative abundance) and Proteobacteria (average 41% relative abundance) at both time points (Appendix A
Appendix A). The 20 most abundant genera across all samples included *Klebsiella*, *Escherichia*/*Shigella*, *Dickeya*, *Haemophilus*, *Pantoea*, *Proteus* and unclassified taxa belonging to the class Gammaproteobacteria (Figure 3A). Abundant Firmicutes genera included *Staphylococcus*, *Enterococcus*, and *Lactobacillus* (class Bacilli), *Clostridium sensu stricto*, *Lachnoclostridium*, *Clostridioides*, and *Finegoldia* (class Clostridia), and *Veillonella* (class Negativicutes; Figure 3A). *Bacteroides* (phylum Bacteroidetes and class Bacteroidia) and *Actinomyces* (phylum Actinobacteria and class Actinobacteria) were the ninth and 20th most abundant genera across samples, respectively (Figure 3A). Twenty-three OTUs were shared across all cohorts and time points, constituting the core microbiota of preterm infants in this study (Figure 3B). These were classified to the class Actinobacteria (*Varibaculum*, *Corynebacterium*, and *Cutibacterium*), Coriobacteriia (*Eggerthella*), Bacilli (*Gemella*, *Staphylococcus*, and *Enterococcus*), Clostridia (*Anaerococcus* and *Clostridium sensu stricto*), Negativicutes (*Negativicoccus* and *Veillonella*), and Gammaproteobacteria (*Escherichia*-*Shigella* and other unclassified taxa).

Sixty-two OTUs were unique to the Carle NICU, while 131 were unique to the TGH NICU (Figure 3B). LDA Effect Size (LEfSe) using NICU (Carle vs. TGH) as class and sample type (multiplet vs. singleton) as subclass showed that *Staphylococcus* and its associated family, order, class, and phyla (*Staphylococcaceae*/Bacillales/Bacilli/Firmicutes) were enriched in TGH samples on/before three weeks postnatal age (Figure 3C). After three weeks postnatal age, *Enterococcus*, *Lactobacillaceae* and their associated order, class, and phyla (Bacillales/Bacilli/Firmicutes) were enriched in TGH relative to Carle samples. Gut microbial abundances were significantly influenced by weeks on BMF, current weight, and weight gain velocity in Carle singletons, TGH multiplets, and TGH singletons (Figure 4). Corrected gestational age was a significant explanatory variable of observed OTU abundances in Carle singletons and TGH multiplets (Figure 4). In Carle multiplet samples, birth weight was the only significant explanatory variable of microbial abundances (Figure 4A).

### 3.4. Gut Microbiota Similarities between Siblings

Compared to singletons and genetically unrelated multiplets, beta diversity Aitchison distances [51] were significantly lower between related multiplets, except for Carle samples ≤3 weeks postnatal age (Figure 5A). After residualizing against clinical covariates, 30 OTUs were commonly correlated between siblings from both Carle and TGH NICUs before and after 3 weeks postnatal age (Figure 5B). These were predominantly (40%; *n* = 15) Gammaproteobacteria, including OTUs classified to the families *Pasteurellaceae* and *Enterobacteriaceae*, and genera *Citrobacter*, *Escherichia*/*Shigella* and *Klebsiella* (Table 3). Other significantly correlated OTUs were classified to the genera *Bifidobacterium* (*n* = 5), *Staphylococcus* (*n* = 2), *Streptococcus* (*n* = 1), *Bacteroides* (*n* = 1), *Anaerococcus* (*n* = 1), *Clostridium sensu stricto* (*n* = 3), (*Clostridium*) *innocuum* group (*n* = 1), and *Veillonella* (*n* = 1; Table 3). In the Carle cohort, one *Corynebacterium*-like OTU was significantly correlated between siblings ≤3 weeks postnatal age, while another *Anaerococcus*-like OTU was significantly correlated between siblings >3 weeks postnatal age (Figure 5B). Meanwhile, in the TGH cohort, 14 OTUs classified to the bacterial taxa *Finegoldia*, *Clostridioides*, *Intestinibacter*, *Clostridium sensu stricto*, *Atopobium*, *Corynebacterium*, *Enterobacteriaceae*, *Klebsiella* (*n* = 3), *Dermabacter*, and *Cutibacterium* (*n* = 2) were significantly correlated between siblings ≤3 weeks postnatal age (Figure 5B). Only three OTUs classified to the genera *Bifidobacterium*, *Streptococcus*, and *Staphylococcus* were significantly correlated between siblings >3 weeks postnatal age from the TGH cohort (Figure 5B).

## 4. Discussion

The preterm infant gut microbiota is dysbiotic [4,5,6,7,8,9,10,11] and perturbed by therapeutic, environmental, and endogenous factors [6,16]. Although clinical and environmental influences on preterm infant gut microbiota are relatively well-studied [6,16], the effects of genetic similarities on gut microbial diversity remain poorly understood. It is currently known that preterm siblings share higher microbiota similarities (lower beta diversity differences) compared to genetically unrelated individuals [12]. However, it is not known whether siblings also share similar alpha diversity (microbial richness and evenness), or whether observed similarities are confounded with clinical variables, such as similar diet, antibiotics exposure, or NICU exposure. In this study, we focused on examining the effects of both clinical variables and genetic relatedness on gut microbial diversity and composition in genetically related and unrelated preterm infants. To address our research objective, we analyzed the clinical data and microbial diversity of stool samples collected before and after 3 weeks postnatal age from the TGH and Carle NICUs.

Weeks on BMF and weight gain velocity were significant predictors of alpha diversity in most samples, after controlling for random effects using linear mixed effects modelling. Consistent with previous studies [4,5,6,7,8,9,10,11], preterm infant gut microbiota in our study were rich in Gammaproteobacteria OTUs, including those classified as *Klebsiella*, *Escherichia*/*Shigella*, *Dickeya*, *Haemophilus*, *Pantoa*, and *Proteus*. The OTU abundances (beta diversity) in most samples were similarly influenced by weeks on BMF, current weight, and weight gain velocity. Our study is one of few to report the effects of BMF on preterm infant gut microbiota, which remain poorly understood [54]. Nevertheless, several studies have cautioned that BMF may increase the risks of feeding tolerance and NEC in preterm infants [55,56]. Thus, connections between BMF feeding, gut microbial diversity, and clinical outcomes in preterm infants require further study. In comparison, the observed effects of weight gain on preterm infant gut microbial diversity were consistent with our previous longitudinal study focusing on a larger TGH cohort [22]. Birth weight also influenced OTU abundances in the Carle multiplet samples. However, because this effect was not consistently observed across all subgroups, the influence of birth weight may be due to random or sampling effects.

Contrary to other studies [22,23], our study observed minimal effects of mother’s own milk on the preterm infant gut microbiome. Previous studies, including ours, reported positive correlations of mother’s own milk with microbial alpha diversity [22,23]. Similarly, we observed significant negative correlations between alpha diversity and percentage of donor human milk, which in turn was negatively correlated with percentage of mother’s own milk. However, the effects of donor human milk were non-significant after modelling for random effects. Nevertheless, we identified inter-NICU differences that could be due to different milk and milk fortifier feeding practices. Compared to Carle NICU samples, TGH NICU samples ≤3 weeks postnatal age were enriched in potentially pathogenic *Staphylococcus*, known to contribute to neonatal sepsis [57]. Interestingly, TGH cohort had 20% rate of culture-positive sepsis versus none in the Carle cohort. After 3 weeks postnatal age, potentially pathogenic *Enterococcus* [58] and *Lactobacillaceae* were enriched in the TGH compared to Carle NICU. Overall, the TGH NICU showed greater abundance of OTUs from the order Bacillales, class Bacilli, and phylum Firmicutes across both time points. These inter-NICU differences may be related to different feeding practices, where infants born <28 weeks from the Carle NICU, received exclusive human milk diet, while those from the TGH NICU were exposed to bovine products from BMF. Lower abundances of *Veillonella* belonging to the phylum Firmicutes were similarly observed in the gut microbiota of preterm infants after 2 to 4 weeks of enteral feeding with human milk and HMF, while higher abundances of another Firmicutes genus, *Terrisporobacter*, were observed in those fed exclusively with formula [21]. NICU-related differences can also be confounded by different infection control practices [59] or different library preparation and sequencing strategies [60]. Future hypothesis-driven studies will be useful in examining the specific effects of HMF on preterm infant gut microbiota composition and diversity. Unlike our previous findings [22] and other studies [12,20], we did not observe significant effects of antibiotic use on preterm infant gut microbial diversity. Delivery mode [11,19] also did not significantly influence gut microbial diversity, presumably because >80% infants in our study were delivered through cesarean section.

Besides clinical and environmental effects, we also evaluated the effects of genetic relatedness on preterm infant gut microbial diversity. Because genetic testing was not performed on infants enrolled in this study, the effects of zygosity on gut microbial diversity, which have been reported in healthy adult [30,61,62] and infant [27,31] populations, were not assessed in this study. As hypothesized, we observed significantly correlated alpha diversity between siblings, especially Faith’s phylogenetic diversity and in the TGH cohort, after residualizing against clinical covariates.

Although we report likely genetic influences on the preterm gut microbiota, we note that genetic effects can be confounded by shared environmental influences, such as maternal [19] or NICU inoculation [7,24]. Higher shared maternal and/or environmental influences possibly explain the higher consistency in alpha diversity correlations between siblings ≤3 weeks compared to >3 weeks postnatal age. Shannon’s diversity values were significantly correlated between siblings in TGH, but not the Carle NICU at both timepoints, which may reflect inter-NICU differences in spatial layout, or in other feeding, clinical, or environmental practices. Specifically, the open bay unit configuration in the Carle NICU likely results in increased human traffic and thus increased diversity of transmissible bacteria [63], compared to the private rooms in the TGH NICU. Consistent with previous analyses of preterm twin infant microbiota [12,32], gut microbial beta diversity profiles in siblings were more similar in siblings compared to unrelated individuals. We also identified 30 candidate “heritable” OTUs belonging to bacterial classes Actinobacteria, Bacilli, Bacteroidia, Clostridia, Erysipelotrichia, and Negativicutes that were commonly correlated between siblings across time points and NICUs. In the future, longitudinal studies on preterm infant multiplets will be instrumental in tracking bacterial diversity metrics and taxa that remain similar between siblings over time [64], and in linking gut microbial diversity variations to differing health outcomes. Metagenomics approaches, which offer improved taxonomic resolution [65], will also be useful in identifying specific bacterial species and strains shared between prematurely born siblings. Additionally, multi-omics studies can further identify bacterial pathways and host immune-related functions that shape similarities and differences in preterm gut infant microbiota.

Despite limitations pertaining to heterogeneity, multicenter collaborations have the potential to address important clinical research questions with improved replication and results validation [66]. Our two-center study represents an effort to deepen the investigation into potential clinical and genetic determinants of preterm infant gut microbiota, with the goal of improving clinical care and the health of preterm neonates. Our findings highlight the need for future integrated and hypothesis-driven studies to disentangle the complex interactions between the preterm infant gut microbiota, maternal and NICU environment, diet, and genetics.

## Figures and Tables

**Figure 1 microorganisms-09-00278-f001:**
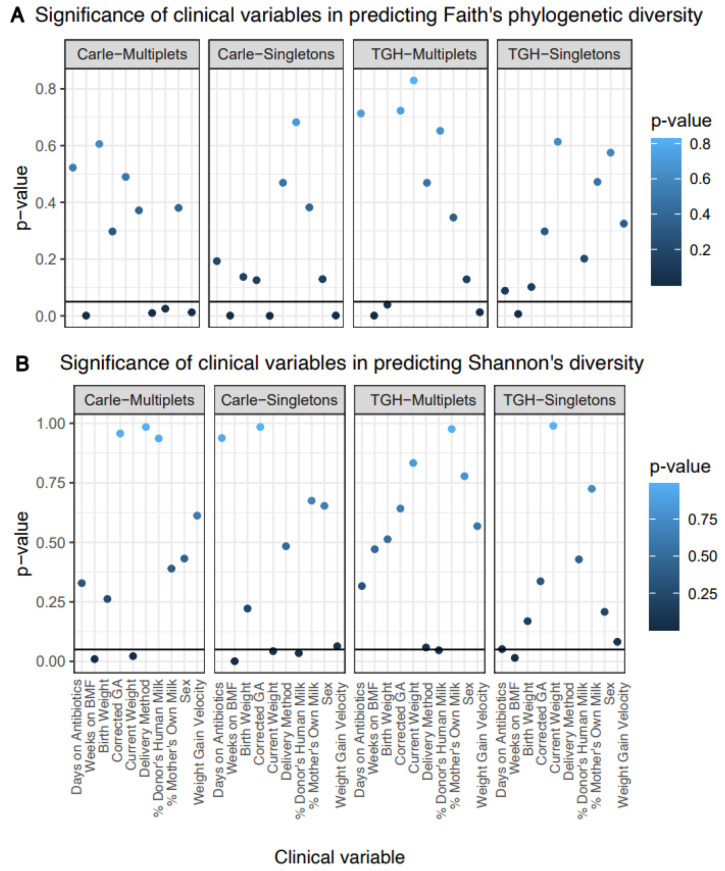
Significance of clinical variables in predicting (**A**) Faith’s phylogenetic diversity and (**B**) Shannon’s diversity, based on ANOVA testing on fitted linear mixed effects models. A horizontal line was plotted at *p* = 0.05 for each subplot. Abbreviations: TGH, Tampa General Hospital; BMF, bovine milk fortifier; GA, gestational age.

**Figure 2 microorganisms-09-00278-f002:**
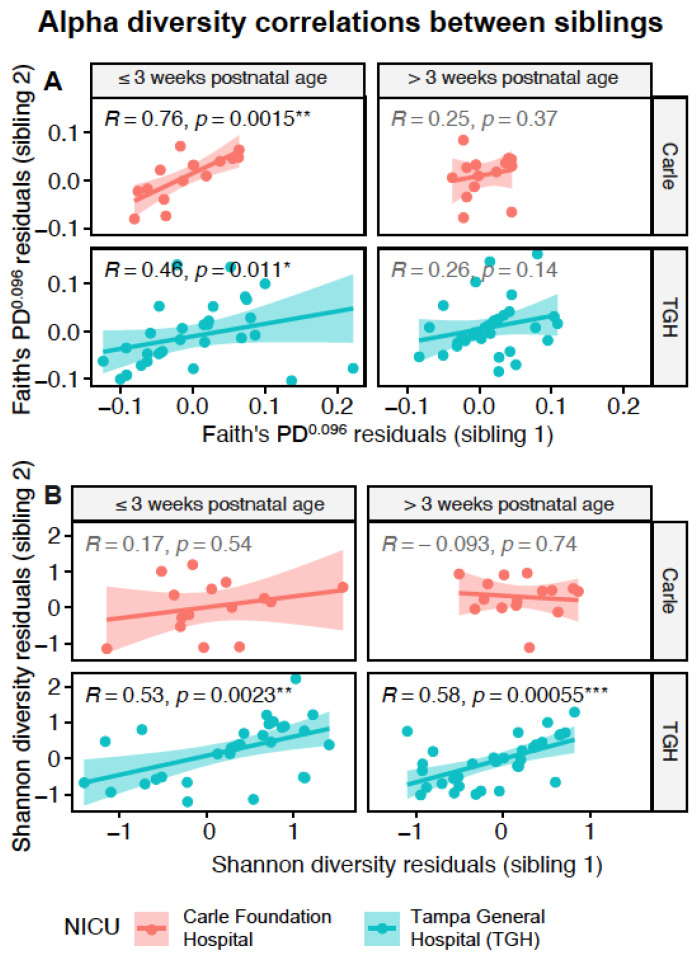
Pairwise correlations of residuals of (**A**) transformed Faith’s phylogenetic diversity and (**B**) Shannon’s diversity between siblings from both Neonatal Intensive Care Units (NICUs), after regressing out clinical variables. Statistically significant comparisons are indicated with * (*p* < 0.05), ** (*p* < 0.01), or *** (*p* < 0.001).

**Figure 3 microorganisms-09-00278-f003:**
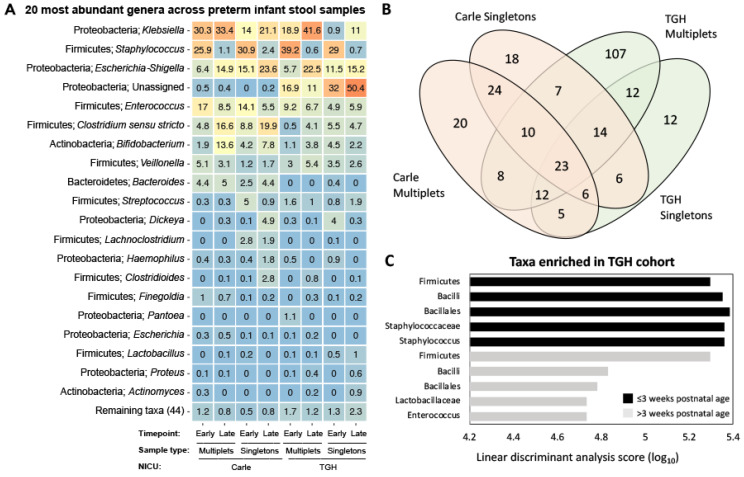
(**A**) Heatmap showing the 20 most abundant genera, and their corresponding phyla, across all samples. Cell values represent mean read abundances across samples; (**B**) Venn diagram showing shared operational taxonomic units (OTUs) among sample groups; and (**C**) histogram of the linear discriminant analysis score calculated for enriched taxa in TGH compared to Carle samples, before and after three weeks postnatal age. The score for each taxon indicates the extent of consistent relative abundance difference between Carle and TGH samples.

**Figure 4 microorganisms-09-00278-f004:**
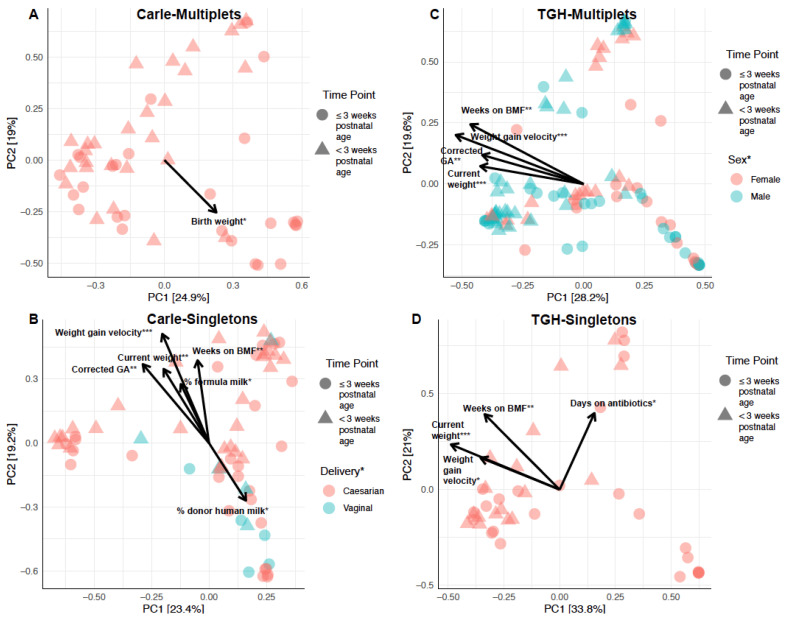
Principal components analysis (PCA) biplot showing the effects of clinical variables on sample clustering in (**A**) Carle multiplets; (**B**) Carle singletons; (**C**) Tampa General Hospital (TGH) multiplets; and (**D**) TGH singletons. Abbreviations: BMF, bovine milk fortifier; GA, gestational age. Statistically significant variables are indicated with * (*p* < 0.05), ** (*p* < 0.01), or *** (*p* < 0.001).

**Figure 5 microorganisms-09-00278-f005:**
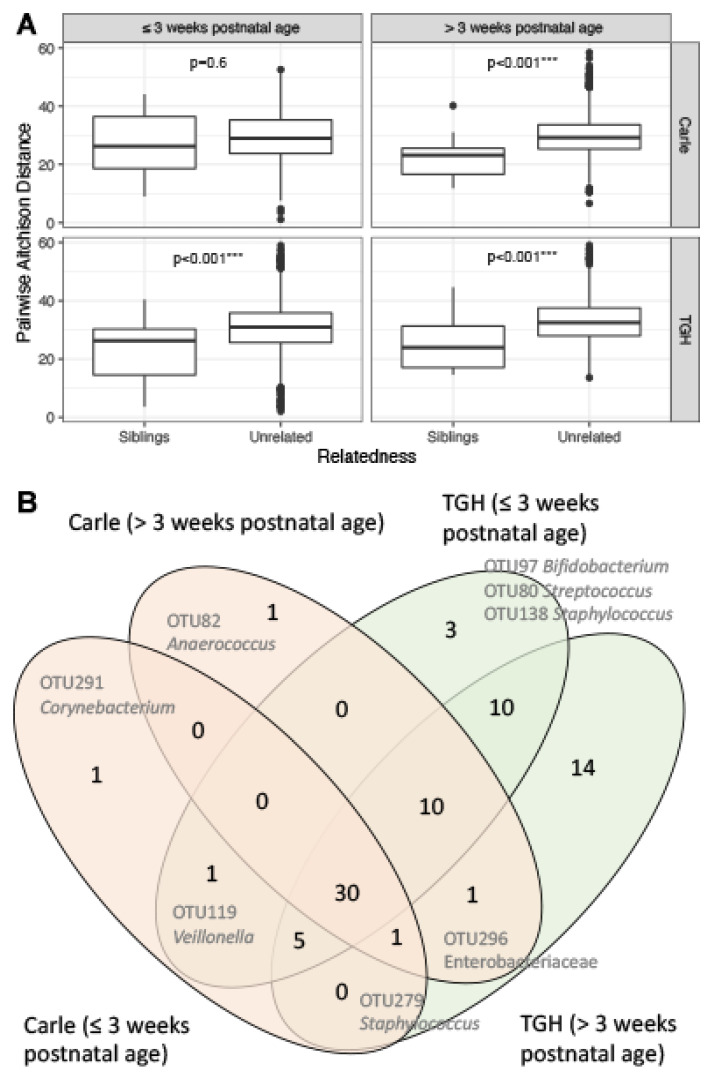
(**A**) Pairwise Aitchison distances between siblings and unrelated individuals and (**B**) Venn diagram showing the numbers of significantly correlated OTUs between siblings after residualizing against clinical variables. Statistically significant comparisons in (**A**) are indicated with *** (*p* < 0.001). Identifiers and taxonomic names of the 30 commonly correlated OTUs between siblings in (**B**) are listed in Table 2. Abbreviation: TGH, Tampa General Hospital.

**Table 1 microorganisms-09-00278-t001:** Demographics and clinical characteristics of the population at birth.

Carle Hospital *n* = 118 (44.7)				Tampa General Hospital *n* = 146 (55.3)			
Cohort Type, *n*, (%)	Multiplet *n* = 54 (45.8)	Singleton *n* = 64 (54.2)	*p* ^1^	Multiplet *n* = 105 (71.9)	Singleton *n* = 41 (28.1)	*p* ^2^	*p* ^3^
Gestational age at delivery, weeks	28.8 (2.3)	28.9 (2.2)	0.89	29.4 (2.2)	28.5 (2.7)	**0.001**	0.12
Birth weight, grams	1246.9 (443.1)	1238.7 (399.2)	0.93	1349.4 (390.5)	1118.7 (199.3)	**0.0004**	0.24
Sex Female, *n*, (%) Male, *n*, (%)	30 (55.6) 24 (44.4)	42 (65.6) 22 (34.4)	0.35	42 (40.0) 63 (60.0)	15 (36.6) 26 (63.4)	0.84	**0.0006**
Type of delivery Vaginal delivery, *n*, (%) Cesarean delivery, *n*, (%)	4 (7.4) 50 (92.6)	10 (15.6) 54 (84.4)	0.27	4 (3.8) 101 (96.2)	- 41 (100.0)	0.48	**0.007**
Number of placentas Monochorionic twins, *n*, (%) Dichorionic twins, *n*, (%)	12 (22.2) 42 (77.8)	- -	-	24 (22.9) 81 (77.1)	- -	-	0.9

Data expressed as means (SD) unless otherwise noted. *p*
^1^: Comparisons between multiplet and singleton groups of Carle Hospital. *p*
^2^: Comparisons between multiplet and singleton groups of Tampa General Hospital. *p*
^3^: Comparisons between Carle Hospital and Tampa General Hospital. All statistical comparisons were based on the Kruskal–Wallis test for numerical data and the chi-square test for categorical data. Bold values indicate *p* < 0.05.

**Table 2 microorganisms-09-00278-t002:** Demographics and clinical characteristics of the population at the time of stool collection.

Carle Hospital *n* = 118 (44.7%)					Tampa General Hospital *n* = 146 (55.3%)				
Cohort Type, *n*, (%)	Multiplet *n* = 54 (45.8%)		Singleton *n* = 64 (54.2%)		Multiplet *n* = 105 (71.9%)		Singleton *n* = 41 (28.1%)		*p* *
	Early	Late	Early	Late	Early	Late	Early	Late	
Corrected gestational age, weeks	30.8 (2.0)	33.2 (2.2)	30.6 (2.1)	33.1 (2.1)	31.1 (2.1)	33.9 (1.9)	30.4 (2.4)	33.0 (2.5)	0.31
Postnatal age, days	13.7 (8.2)	29.9 (9.0)	11.8 (6.6)	29.0 (8.0)	10.1 (4.9)	29.2 (7.3)	13.0 (6.4)	31.4 (5.0)	0.62
Current weight, grams	1388.8 (441.4)	1853.1 (597.9)	1369.3 (438.2)	1866.1 (574.2)	1358.2 (376.7)	1877.1 (463.5)	1194.0 (283.5)	1655.7 (260.7)	0.5
Weight gain velocity, g/d	9.3 (9.7)	20.9 (9.1)	6.9 (14.0)	21.2 (10.0)	−1.4 (12.3)	17.8 (6.7)	1.1 (15.5)	17.1 (4.0)	**0.001**
Days on antibiotics, days	0.5 (1.5)	0.2 (1.0)	0.7 (1.5)	0.0 (0.2)	2.7 (2.7)	3.3 (3.7)	4.2 (3.6)	4.5 (3.7)	**<0.0001**
Bovine milk fortifier, weeks	1.4 (1.0)	3.3 (1.8)	1.1 (1.0)	2.8 (2.1)	0.9 (0.8)	3.7 (1.1)	1.5 (1.0)	3.9 (0.8)	0.16
Proportion of mother’s own milk	0.83 (0.28)	0.77 (0.32)	0.63 (0.37)	0.56 (0.46)	0.47 (0.42)	0.53 (0.43)	0.86 (0.29)	0.81 (0.36)	0.15
Proportion of donor’s human milk	0.16 (0.24)	0.13 (0.26)	0.36 (0.38)	0.20 (0.34)	0.34 (0.42)	0.15 (0.30)	0.11 (0.28)	0.00 (0.00)	0.31
Proportion of formula	0.01 (0.07)	0.10 (0.16)	0.01 (0.07)	0.24 (0.36)	0.19 (0.37)	0.32 (0.40)	0.04 (0.09)	0.19 (0.36)	**0.005**

Data expressed as means (SD). Weight gain velocity is calculated as (current weight at stool collection—birth weight)/postnatal day of life at stool collection. The proportion of milk received is based on volume of intake. “Early” stool samples were collected ≤3 weeks postnatal age, while “Late” samples were collected >3 weeks postnatal age. *p* *: Kruskal–Wallis comparisons between Carle Hospital (all samples) and Tampa General Hospital (all samples). Bold values indicate *p* < 0.01.

**Table 3 microorganisms-09-00278-t003:** Taxonomic classification of OTUs observed to be commonly correlated between siblings.

OTU	Class	Family/Genus
OTU190	Actinobacteria	*Bifidobacterium*
OTU221	Actinobacteria	*Bifidobacterium*
OTU210	Actinobacteria	*Bifidobacterium*
OTU147	Actinobacteria	*Bifidobacterium*
OTU52	Actinobacteria	*Bifidobacterium*
OTU233	Bacilli	*Staphylococcus*
OTU117	Bacilli	*Staphylococcus*
OTU176	Bacilli	*Streptococcus*
OTU6	Bacteroidia	*Bacteroides*
OTU200	Clostridia	*Anaerococcus*
OTU178	Clostridia	*Clostridium sensu stricto*
OTU28	Clostridia	*Clostridium sensu stricto*
OTU187	Clostridia	*Clostridium sensu stricto*
OTU45	Erysipelotrichia	(*Clostridium*) *innocuum* group
OTU283	Gammaproteobacteria	Pasteurellaceae
OTU284	Gammaproteobacteria	Enterobacteriaceae
OTU195	Gammaproteobacteria	Enterobacteriaceae
OTU141	Gammaproteobacteria	Enterobacteriaceae
OTU162	Gammaproteobacteria	Enterobacteriaceae
OTU222	Gammaproteobacteria	Enterobacteriaceae
OTU173	Gammaproteobacteria	Enterobacteriaceae
OTU58	Gammaproteobacteria	*Citrobacter*
OTU246	Gammaproteobacteria	*Citrobacter*
OTU272	Gammaproteobacteria	*Escherichia*
OTU234	Gammaproteobacteria	*Escherichia*
OTU105	Gammaproteobacteria	*Escherichia-Shigella*
OTU148	Gammaproteobacteria	*Escherichia-Shigella*
OTU24	Gammaproteobacteria	*Escherichia-Shigella*
OTU10	Gammaproteobacteria	*Klebsiella*
OTU133	Negativicutes	*Veillonella*

## Data Availability

The data (sequenced reads) presented in this study are publicly available on National Center for Biotechnology Information’s (NCBI) Sequence Read Archive (SRA) under the BioProject accessions PRJNA449987 and PRJNA693089.

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
