# Peer review of "The Effects of Genetic Relatedness on the Preterm Infant Gut Microbiota"

_microorganisms, 2021, doi:10.3390/microorganisms9020278_

Round 1

Reviewer 1 Report

In this manuscript, the authors described the preterm gut infant microbiota from singletons and twins using the alpha and beta-diversity parameters. The study is well conceptualized and the samples were retrieved from two different Neonatal Intensive Care Units (NICUs).
Minor comments:
- Abstract, line 21, please for the first time in the manuscript is better that “Neonatal Intensive Care Units” is written with uppercase letters.
- Introduction, line 50, could the authors change the citation 19 with a more appropriate citation?
- Materials and Methods, line 114, could the authors insert a supplementary table with the sequence of the primers used in this study?
- Materials and Methods, line 162 and all the manuscript, please could the authors substitute PCA with PCoA?
- Results, lines 178-179 and table 1, the numbers are different between the main text and the table. Could the authors explain better the sample numbers with a further table?
- Results, lines 180-181, are the differences between the two cohorts significant? If yes specify it.
- Results, lines 183-187. Are the samples cited included in the analysis? If yes it is more appropriate, eliminate them.
- Results, lines 232-233, could the authors insert the average percentage of the dominated phyla retrieved in the samples?
- Figure 3a. Maybe, it is better add to the manuscript a supplementary table with all the percentage of the phyla for each sample analyzed.

Author Response

In this manuscript, the authors described the preterm gut infant microbiota from singletons and twins using the alpha and beta-diversity parameters. The study is well conceptualized and the samples were retrieved from two different Neonatal Intensive Care Units (NICUs).

We thank the reviewer for the encouraging feedback.

Minor comments:
- Abstract, line 21, please for the first time in the manuscript is better that “Neonatal Intensive Care Units” is written with uppercase letters.

We fixed this in line 21 of the revised manuscript.

- Introduction, line 50, could the authors change the citation 19 with a more appropriate citation?

We did not understand why this citation was not appropriate and would prefer to keep citation 19. In the paper (doi:10.1016/j.jpeds.2014.09.041), preterm infants whose mothers who received intrapartum antimicrobial prophylaxis (IAP) did show altered gut microbiota, evident in Figure 3 of the article and the excerpt below:

“However, at 30 days of age, several statistically significant differences were observed, with the infants not exposed to antibiotics (either directly or via their mothers) having higher relative amounts of Comamonadaceae, Staphylococcaceae, and unclassified Bacilli than the other 3 groups (P < .05). At this time point, infants not exposed to antibiotics also had significantly higher percentages (P < .05) of Bifidobacteriaceae, Streptococcaceae, unclassified Actinobacteria, and unclassified Lactobacillales and lower (P < .05) of Enterobacteriaceae than both groups of infants whose mothers received IAP (independently on whether or not the infant received antibiotics).”

- Materials and Methods, line 114, could the authors insert a supplementary table with the sequence of the primers used in this study?

 As requested, we added Table S1 containing primer sequences used in the study.

- Materials and Methods, line 162 and all the manuscript, please could the authors substitute PCA with PCoA?

The plots presented in this manuscript are PCA (Principal Components Analysis), not PCoA (Principal Coordinates Analysis) plots. We used PCA instead of PCoA because PCA uses Euclidean Distance on Hellinger-transformed data and does not require an explicit distance measure.

- Results, lines 178-179 and table 1, the numbers are different between the main text and the table. Could the authors explain better the sample numbers with a further table?

Lines 178-179 in the original manuscript shows the total number of twin/triplet sets and singleton individuals analyzed for this study, while Table 1 shows the total number of stool samples (rather than individuals or twin pairs) collected at both early and late time points. We understand that this can be confusing. We clarified the statement accordingly in lines 177-180 of the revised manuscript:

“Eleven twin sets, 2 triplet sets, and 32 singletons were recruited from the Carle NICU, while 22 twin sets, 4 triplet sets, and 24 singletons were recruited from the TGH NICU (Table S2). From these individuals, a total of 264 stool samples were collected before and after three weeks postnatal age for this study (Table 1).”

We also added the relationships between sample IDs, twin/triplet sets, and individuals in Table S2 along with DADA2 denoising stats and phylum relative abundances for each sample.

- Results, lines 180-181, are the differences between the two cohorts significant? If yes specify it.

 Yes - the differences between the two cohorts were significant. We clarified the sentence accordingly (lines 181-183; emphasis in bold):

“Overall, the TGH cohort had a significantly higher proportion of male infants, and a significantly higher rate of cesarean delivery (Table 1).”

- Results, lines 183-187. Are the samples cited included in the analysis? If yes it is more appropriate, eliminate them.

The samples referred to by the reviewer were the seven singletons from the TGH NICU that were missing samples from >3 weeks postnatal age and three twin sets from the TGH that were missing samples from ≤ 3 weeks postnatal age. These were included in the analysis and we did not think it necessary to eliminate these samples. This is because these samples 1) represent only a small proportion of our dataset; 2) still contain useful clinical and microbial diversity information (discarding them will be wasteful). Furthermore, in our analyses, we either split the samples into >3 weeks postnatal age and ≤ 3 weeks postnatal age categories or included a variable of time (corrected gestational age). Therefore, missing samples from either time point should not affect these results.

- Results, lines 232-233, could the authors insert the average percentage of the dominated phyla retrieved in the samples?

We thank the reviewer for the suggestion and inserted the average percentages of the dominating phyla (lines 236-238; emphasis in bold):

“Stool samples from both Carle and TGH NICUs were dominated by the phyla Firmicutes (average 52% relative abundance) and Proteobacteria (average 41% relative abundance) at both time points (Table S2).”

- Figure 3a. Maybe, it is better add to the manuscript a supplementary table with all the percentage of the phyla for each sample analyzed.

We added this information to Table S2.

Reviewer 2 Report

L128-129 I would recommend adding a rarefaction graph in order to understand why a cutoff of 10 OTU has been used here. DADA2 denoising stats should also be added as a supplementary table. 

Line 217-218
Please describe which groups have been used for the Kruskal-Wallis comparisons between Carle Foundation Hospital and Tampa General Hospital.
I can see only one p-value and I consequently supposed that only two groups have been used.

Figure 1 - It is not clear to me why "days on antibiotics" with a significant p-value (it's on the 0.05 threshold)  relative to in TGH singleton group are not present among the discussed points, although they emerged in predicting the Shannon diversity. Is the p-value not statistically significant? 

Table 3 - Please move this table to the result section. I cannot see the reason why insert this table into the discussion.

Figure 4 - Please provide in the discussion a reason for the only significant variable "Birth weight" describing the PCA graph relative to Carle-Multiples group.

Figure 5. Please move this figure to the result section.

Author Response

L128-129 I would recommend adding a rarefaction graph in order to understand why a cutoff of 10 OTU has been used here. DADA2 denoising stats should also be added as a supplementary table. 

We thank the reviewer for these recommendations. The cutoff we used was “OTUs with <10 total frequency” instead of 10 OTUs. We added the histogram distribution of OTU total frequency, as well as the rarefaction curves for each stool sample, to Figure S2. We also added DADA2 denoising stats to Table S2.

Line 217-218 
Please describe which groups have been used for the Kruskal-Wallis comparisons between Carle Foundation Hospital and Tampa General Hospital.
I can see only one p-value and I consequently supposed that only two groups have been used. 

Yes. We combined all Carle Foundation Hospital samples and all Tampa General Hospital samples for the Kruskal-Wallis comparisons, so only two groups (Carle Foundation Hospital vs Tampa General Hospital) were used. We clarified the footnote of Table 2 accordingly (emphasis in bold):

“p*: Kruskal-Wallis comparisons between Carle Foundation Hospital (all samples) and Tampa General Hospital (all samples).”

Figure 1 - It is not clear to me why "days on antibiotics" with a significant p-value (it's on the 0.05 threshold) relative to in TGH singleton group are not present among the discussed points, although they emerged in predicting the Shannon diversity. Is the p-value not statistically significant? 

For days on antibiotics, the p-value was slightly above the threshold at p=0.0516 in predicting Shannon Diversity. We added a brief statement on this in lines 205-206 of the revised manuscript:

“Days on antibiotics also predicted Shannon’s diversity in TGH singletons, although the p-value was slightly above the significance threshold (p=0.052; Figure 1).”

Table 3 - Please move this table to the result section. I cannot see the reason why insert this table into the discussion.

We apologize for this and moved Table 3 to the result section.

Figure 4 - Please provide in the discussion a reason for the only significant variable "Birth weight" describing the PCA graph relative to Carle-Multiples group.

We discussed a possible reason in lines 328-331 of the revised manuscript:

“Birth weight also influenced OTU abundances in the Carle multiplet samples. However, because this effect was not consistently observed across all subgroups, the influence of birth weight may be due to random or sampling effects.”

Figure 5. Please move this figure to the result section.

We moved Figure 5 to the result section.

Round 2

Reviewer 1 Report

The Paper is well written and ready for the publication.